# OpenReview forum: "Backdoor Attacks in Federated Learning by Poisoned Word Embeddings"
_aclweb.org/ACL/2022/Workshop/FL4NLP — FL4NLP@ACL2022_

### Official Review · Reviewer_YEBb · 2022-03-21

**Rating:** 5
**Confidence:** 5

**Review:**

The paper presents a new attack extension of the embedding attack on NLP models to FL. Instead of training to optimize the whole model the attacker only focuses on a small single embedding of an unpopular token.

I really liked the idea and think that it has a good potential impact, however I have a couple of concerns:

1. Motivation -- FL in NLP is motivated by a smart keyboard application and therefore language generation task. I did not understand motivation under seq2seq tasks, neither summarization nor translation seem like would be good candidates for FL as there are no privacy constraints. I can understand classification, but not on the news dataset (which is hardly private) but rather some toxicity dataset.

2. Experiments -- some details on seq2seq task would be great otherwise it's unclear what task exactly gets evaluated (I assume it's a summarization task as it uses ROUGE but still not clear). "Trigger range" discussion is also complex as it wasn't introduced before.

3. Novelty -- the backdoor attacks on embeddings exist in literature as well as backdoor attacks on FL. Seems like it's a trivial operation to apply one to another. I cannot see why 3.3 is novel as it's the core assumption in all other backdoor FL papers -- other participants contributions can be ignored when computing backdoored model update.

In my opinion the key interesting part of the paper is that it can possibly evade norm-bound detection by modifying only a small model's embedding vector, however it has a very trivial way to defend -- simply check for norm updates of each embedding vector.

Overall, I really like the idea but it needs more solid motivation and exploration.

---

### Official Review · Reviewer_ECVk · 2022-03-22
**Interesting topic and the method can be further improved**

**Rating:** 5
**Confidence:** 3

**Review:**

This work proposes a practical backdoor attack against NLP models in the Federate Learning scenarios by inserting malicious tokens in the word embeddings. The author demonstrates its effectiveness through many practical scenarios, e.g., large trigger tokens, etc. It would be better to explore inserting backdoor triggers in a more stealth manner, e.g., inserting incontinuous or dynamic backdoor triggers, inserting backdoor triggers without adding too many tokens besides the benign tokens. Since the attack method is adapted from CV methods, the robustness of the proposed method against potential defense mechanisms adapted from Image Classification tasks [1,2] should also be discussed.

[1] Wang, Bolun, et al. "Neural cleanse: Identifying and mitigating backdoor attacks in neural networks." 2019 IEEE Symposium on Security and Privacy (SP). IEEE, 2019.

[2] Guo, Junfeng, Ang Li, and Cong Liu. "AEVA: Black-box Backdoor Detection Using Adversarial Extreme Value Analysis." ICLR (2022).

---

### Official Review · Reviewer_fc7i · 2022-03-24
**Effective method for conducting backdoor attack of federated NLP tasks**

**Rating:** 7
**Confidence:** 4

**Review:**

This paper introduces a practical approach for injecting backdoor attacks into a federated learned model. The attackers only manipulate the embedding layers of a model for injecting the backdoor. Compared to the previously proposed backdoor attack on language models (where the attacker manipulates to change all layers' weights), the proposed attack is easier to inject and is harder to detect by the central server.

Extensive experimental results indicate that the proposed attack is effective under various NLP tasks and transformer models. I'm convinced that the proposed attack is effective given the scales of the experiments.

---

### Decision · Program_Chairs · 2022-03-26

Accept